# Diagnostic of Patients with COVID-19 Pneumonia Using Passive Medical Microwave Radiometry (MWR)

**DOI:** 10.3390/diagnostics13152585

**Published:** 2023-08-03

**Authors:** Berik Emilov, Aleksander Sorokin, Meder Seiitov, Binsei Toshi Kobayashi, Tulegen Chubakov, Sergey Vesnin, Illarion Popov, Aleksandra Krylova, Igor Goryanin

**Affiliations:** 1Educational-Scientific Medical Center, Kyrgyz State Medical Academy Named after Isa Akhunbaev, Bishkek 720040, Kyrgyzstan; 2Department of Physics, Medical Informatics and Biology, Kyrgyz-Russian Slavic University Named after Boris Yeltsin, Bishkek 720000, Kyrgyzstan; aasorokin@rambler.ru; 3Well Being Ginza, Tokyo 104-0061, Japan; binsei.nd@gmail.com; 4Kyrgyz State Medical Institute of Post-Graduate Training and Continuous Education Named after S.B. Daniyarov, Bishkek 720040, Kyrgyzstan; t_53chubakov@mail.ru; 5Medical Microwave Radiometry Ltd., Edinburgh EH10 5LZ, UK; vesnin47@gmail.com; 6Faculty of Mathematics and Information Technology, Volgograd State University, 400062 Volgograd, Russia; popov.larion@yandex.ru (I.P.); krylova_kae@mail.ru (A.K.); 7School of Informatics, University of Edinburgh, Edinburgh EH8 9AZ, UK; 8Biological Systems Unit, Okinawa Institute Science and Technology, Kunigami District, Okinawa 904-0495, Japan

**Keywords:** 2019-nCoV, COVID-19, RT-PCR, SARS-CoV-2, community-acquired pneumonia, chest CT, microwave radiometry, temperature measurement

## Abstract

Background. Chest CT is widely regarded as a dependable imaging technique for detecting pneumonia in COVID-19 patients, but there is growing interest in microwave radiometry (MWR) of the lungs as a possible substitute for diagnosing lung involvement. Aim. The aim of this study is to examine the utility of the MWR approach as a screening tool for diagnosing pneumonia with complications in patients with COVID-19. Methods. Our study involved two groups of participants. The control group consisted of 50 individuals (24 male and 26 female) between the ages of 20 and 70 years who underwent clinical evaluations and had no known medical conditions. The main group included 142 participants (67 men and 75 women) between the ages of 20 and 87 years who were diagnosed with COVID-19 complicated by pneumonia and were admitted to the emergency department between June 2020 to June 2021. Skin and lung temperatures were measured at 14 points, including 2 additional reference points, using a previously established method. Lung temperature data were obtained with the MWR2020 (MMWR LTD, Edinburgh, UK). All participants underwent clinical evaluations, laboratory tests, chest CT scans, MWR of the lungs, and reverse transcriptase polymerase chain reaction (RT-PCR) testing for SARS-CoV-2. Results. The MWR exhibits a high predictive capacity as demonstrated by its sensitivity of 97.6% and specificity of 92.7%. Conclusions. MWR of the lungs can be a valuable substitute for chest CT in diagnosing pneumonia in patients with COVID-19, especially in situations where chest CT is unavailable or impractical.

## 1. Introduction

In December 2019, a group of patients with pneumonia of unknown origin was found in Wuhan, China [1]. In the weeks that followed, this unknown virus gradually spread around the world [2]. In January 2020, a research institution in China announced that viral pneumonia is a new coronavirus—SARS-CoV-2, and the World Health Organization (WHO) later named the disease COVID-19 [3]. According to Worldometer data, at the end of August 2020, 25,416,807 cases were confirmed, of which 851,102 people died and 17,724,602 people recovered worldwide [4]. Mortality from COVID-19 is lower than from severe acute respiratory syndrome (SARS) and Middle East respiratory syndrome (MERS) [5,6,7]. However, COVID-19 is extremely transmissible and spread rapidly, as it can be transmitted by airborne droplets and contact [8]. The incubation period of COVID-19 infection most often lasts from 1 to 14 days and can last up to 24 days, with most cases showing symptoms within 5-6 days after infection. However, it is important to note that some people may be asymptomatic and can still spread the virus to others [9,10,11]. To date, there is no evidence of any effective treatments for patients with COVID-19 [12] and due to the lack of specific drugs for COVID-19, it is important to forecast the disease outbreak in order to determine the load on medical institutions and to detect and manage the disease at an early stage to immediately isolate people who are confirmed to be infected with COVID-19 [13,14,15,16,17,18,19]. Real-time reverse transcription polymerase chain reaction (RT-PCR) analysis is considered to be the first tool in the diagnosis of COVID-19 [20].

The primary symptoms of SARS-CoV-2 infection are similar to those of other coronaviruses, such as fever, cough, and fatigue, and tend to present as flu-like symptoms [21]. However, certain high-risk groups, such as the elderly and those with preexisting medical conditions, are at a greater risk of experiencing severe respiratory issues, including acute respiratory distress syndrome, interstitial pneumonia, and multiple organ failure, which may result in varying levels of shortness of breath and specific radiological signs [22,23].

Bilateral interstitial infiltration is the most prevalent manifestation of COVID-19 in the lungs, which can significantly disrupt the balance between ventilation and perfusion [15,24]. In severe cases, this can lead to respiratory failure and even death. While chest computed tomography (CT) has been the preferred diagnostic and monitoring tool for COVID-19, logistical challenges associated with patient transportation and disinfection of CT rooms have posed significant challenges. Moreover, chest CT is not readily available in many low- and middle-income countries, further limiting its utility as a diagnostic tool [25,26].

In addition, in recent studies, attention has been drawn to the importance of chest CT, since in examined COVID-19 patients with a false-negative RT-PCR result [1,27,28], the sensitivity of chest CT was 98% and chest CT is of great importance not only for the diagnosis of COVID-19, but also for monitoring the progression and severity of the disease and assessing the therapeutic effect.

Diagnostic chest CT can be used to monitor the extent of lung involvement and monitor any changes in patients whose RT-PCR tests for SARS-CoV-2 and plain chest radiographs were negative [15,29,30,31]. Several studies have reported that chest CT shows typical imaging features in almost all patients with COVID-19 [32,33,34,35] Hung et al. and Xie et al. (2020) emphasize that similar imaging features were also found in patients whose RT-PCR tests were negative [1,27].

However, due to the problems of infection control associated with the transportation of the patient, as well as the disinfection of chest CT rooms after examining the patient and the lack of availability of chest CT, created some kind of obstacles [20,36].

Furthermore, in mountainous countries such as Kyrgyzstan, chest CT is not always available for the diagnosis of pneumonia, and so researchers have proposed using passive microwave radiometry (MWR) as an alternative diagnostic method, which was previously used in practical medicine [37,38]. Infrared thermometry is a method for determining skin temperature based on measuring tissue radiation in the infrared wavelength range. MWR is a method for measuring the integral deep temperature of human internal tissues based on measuring the power of the tissue’s own thermal radiation in the microwave wavelength range. The dual band sensor with a built-in antenna for simultaneous measurement of core temperature and skin temperature. It is a safe and side-effect-free technique, as it only measures the patient’s radio-thermal radiation. The system is accurate to within ±0.2 °C and can detect temperature differences in soft tissues at depths of 3 to 7 cm. The measurement area is usually 3–5 cm in diameter [39]. While MWR is not yet widely used, it has the potential to be an effective substitute for chest CT, particularly in regions where CT is not easily accessible. We study the experience of using the MWR method as a diagnostic screening in patients with COVID-19 complicated pneumonia.

The rationale for the use of MWR is the rise of tissue temperature due to the inflammatory process. The local alterations in the synthesis and release of immune mediators in response to injury result in altered biochemical processes and temperature local changes.

Indeed, existing evidence suggests that local inflammation increased temperature that can be detected by the MWR sensor, which is placed on the skin over the inflamed area [40].

The ability of MWR to detect with high accuracy in-depth temperature changes in human tissues is under investigation in various medical fields. There are numerous clinical applications for non-invasive monitoring of deep tissue temperature. We present the design and experimental performance of a MWR measuring the volume temperature of tissue regions, especially the lungs. The local temperature changes precede structural changes, it is of paramount importance to diagnose inflammatory changes as early as possible, even if signs and symptoms suggestive of inflammation are absent.

## 2. Materials and Methods

The aim of the study is to examine the utility of the MWR approach as a screening tool for diagnosing pneumonia with complications in patients with COVID-19. The Bioethics Committee (protocol N4 11 November 2021) approved a study. Before the study, each participant had provided written informed consent.

The researcher who did the measurements was qualified and trained according to international standards (MWR diagnostic method for COVID-19, Edinburgh, UK). There are 30 points on the participant’s chest, 28 of which are symmetrical (R1-R14, L1-L14) and 2 control points (FR and BC), configured to perform these measurements, record temperatures and build the corresponding temperature fields. Most of the patients were naked (90) and a minority were in thin clothing (52) and the procedure took maximum of 10 min [41] The lung and skin temperature data were obtained using the MWR2020 device (formerly RTM-01-RES) MMWR Ltd., Edinburgh, UK. The MWR2020 operates in the frequency range 3400–3900 MHz. The MWR bandwidth is 500 MHz [42].

The principle of operation of a microwave radiometer is a result of the fact that in the microwave range, the radiation power of human tissues is proportional to the temperature of these tissues. Therefore, by measuring the intrinsic radiation of tissues in the microwave frequency range, it is possible to obtain information about the temperature of internal tissues.

The temperature measured by a microwave radiometer is the average temperature of the tissues in the volume below the antenna. An anti-interference antenna with a slot radiator is used as an antenna. The electric field of the antenna is oriented perpendicular to the slot. The presented antenna allows temperature measurements without additional room shielding. The reflectance of the antenna power for different parts of the body and different patients is within R2 = 0 − 0.1. This variation is since the thickness of the fat and muscle layers in different patients can vary significantly. It follows from this that about 10% of the power radiated by human tissues is reflected from the antenna and does not enter the receiving device, which can significantly reduce the accuracy of temperature measurement.

The microwave radiometer compensates for measurement error due to antenna reflections. This is achieved by “noising” the input circuits of the radiometer from the load side of the circulator, which has a temperature close to the measured temperature. Therefore, the microwave radiometer used provides the necessary measurement accuracy in the presence of antenna mismatch.

To ensure safety, strict and standardized procedures were implemented to protect the operators and decontaminate the instruments during and after the procedure [43]. The time difference between lung MWR and chest CT measurements in patients with COVID-19 complicated pneumonia was no more than 48 h.

Patients who did not exhibit clinical presentation of COVID-19, tested negative for RT-PCR for SARS-CoV-2, or had unconfirmed bilateral pneumonia on chest CT were excluded from the main group. The patients from the control group were excluded if any member exhibited symptoms or complaints or tested positive for RT-PCR for SARS-CoV-2.

All participants in the main group underwent clinical examinations with auscultation of the lungs, body temperature measurement, laboratory tests including C-reactive protein (CRP), chest CT, MWR of the lungs and RT-PCR test for SARS-CoV-2. It’s important to note that the medical rooms used for the study were designed according to hygienic standards and microclimatic conditions. The maintenance of consistent humidity and temperature levels in medical rooms is an important consideration, as variations in these factors can potentially impact the accuracy of the measurements taken. By ensuring that these conditions were consistent across all measurements, the researchers were able to minimize this potential source of error. The humidity in the room ranged from 40% to 50% for all measurements, and the room temperature was maintained between 22–25 °C to ensure the accuracy and consistency of the results.

IBM SPSS Statistics, version 26 (Chicago, IL, USA) was used to perform the statistical analysis. Receiver operating characteristics (ROC) curves were employed to determine the most effective classifier for diagnosis, with the area under the curve (AUC) analyzed. The logistic regression model was used as the predictive model, and its sensitivity and specificity were calculated, along with the coefficients of determination of Cox-Snell and Nagelkerke. The Hosmer-Lemeshow test and the Chi-square test were conducted to evaluate the significance of the final model. Additionally, the Pearson correlation coefficient was used to measure the correlation between the predictors.

ROC curves provide an estimate of the overall accuracy of a diagnostic test by plotting the sensitivity against the false positive rate for a range of cut-off values. The AUC is a summary statistic of the ROC curve, with a higher value indicating better diagnostic performance.

Logistic regression models are frequently used to predict the probability of an outcome, such as the presence or absence of a disease. Sensitivity and specificity are important measures for assessing the model’s ability to identify positive and negative cases, while the Cox-Snell and Nagelkerke coefficients of determination provide insight into the proportion of variation in the outcome explained by the predictor variables. The Hosmer-Lemeshow test evaluates the goodness-of-fit of the model, while the Chi-square test assesses its overall significance.

The Pearson correlation coefficient measures the linear relationship between two variables, ranging from −1 to 1, with 0 indicating no correlation. It helps to determine the strength and direction of the relationship between predictor variables and can reveal any potential interactions or dependencies between them.

## 3. Results

The study included 142 patients from the main group who was admitted to the outpatient emergency department of the Medical Center of Kyrgyz State Medical Academy for the pandemic period and diagnosed with COVID-19 complicated pneumonia. The patients’ age ranged from 20 to 87 years, with 67 men and 75 women. A control group of 50 healthy individuals, consisting of 24 men and 26 women, ranging in age from 20 to 70 years, was also included.

All patients underwent chest CT and MWR of the lungs. It should be noted that on chest CT images of the lungs there were extensive areas of infiltration on both sides confirmed by radiologists, more than five years’ experience for chest CT. The MWR scans are able to detect and indicate areas of inflammation using different colors, which indicate fresh areas of inflammation or its absence. The healthy patients who had chest CT were compared by MWR of the lungs examination at all indicated points. The lungs of a healthy person are shown, where red areas of active inflammation are not visible, which is confirmed by a chest CT (Figure 1) and on MWR images of healthy participants, we can see yellow, green areas. Yellow areas may indicate places with higher temperature values, and green areas with relatively low temperature differences. It is important to note that there are no red spots here.

The data obtained from MWR includes variations in temperature between the skin and internal (lung) temperatures.

When comparing chest CT images with MWR of the lungs, there were similar data on the presence of inflammation in the form of ground-glass opacity (GGO) and red areas on the MWR in patients with COVID-19 complicated by pneumonia, which we can see in Figure 2.

Figure 2 shows a MWR difference in temperature between the left and right lung, with the green areas indicating a lower temperature and the red areas indicating a higher temperature due to inflammation. This temperature difference is similar to what is observed in chest CT scans as GGOs and they correspond to CT images. The statement also suggests that temperature differences were observed between healthy individuals and COVID-19 patients with complicated pneumonia. This could indicate that MWR imaging could potentially be used as a diagnostic tool to distinguish between healthy individuals and those with COVID-19 pneumonia.

In this study, 142 participants with pneumonia caused by COVID-19 were found to have lung tissue inflammation confirmed in 140 using MWR, of the control group, the absence of inflammation was confirmed in 42 of 50 healthy participants.

The MWR of the lungs generated a large volume of temperature data, and to evaluate the effectiveness of the MWR method and to identify COVID-19 pneumonia, we employed ROC analysis by measured temperatures. This analysis provides an accuracy calculation of a binary classification by varying a threshold value T in the range of feature X values. If X < T, then the object belongs to the healthy group, otherwise to the pneumonia group.

Table 1 and Table 2 provide descriptive statistics for the mean values across all measurement points for each subject. As can be seen from the tables, there are quite significant differences both between the surface and internal temperatures, and between sick and healthy people. This, in turn, allows us to hope for the possibility of using these parameters for the prognosis of the disease.

Figure 3 illustrates all 60 ROC curves generated based on measuring both external (skin) and internal (lung) temperatures at each of the examined points.

The ROC curves in Figure 3 demonstrate high values of the area under the curve, ranging from 0.816 to 0.956, indicating that temperature values can be used as a predictor for the logistic model. However, it is impractical to work with all 60 predictors, as this approach lacks generality. Thus, an integral variable that can be applied in any case needs to be formulated. To achieve this, average skin and internal temperature values were taken, providing an average over surface points and internal points, respectively. These integral variables offer a more generalizable approach compared to working with individual temperature values at each point.

ROC curves for the average internal and external temperatures are displayed in Figure 4. Based on the plot and the parameters presented in Table 3, it could be concluded that both variables are highly effective predictors.

Both ROC curves exhibit high values of the area under the curve and relatively narrow confidence intervals. The area under the curve for average internal temperature is 0.967 ± 0.013 (95% CI 0.941–0.993), while for average skin temperature it is 0.951 ± 0.016 (95% CI 919–0.983). These findings suggest that both variables are effective predictors. Consequently, a logistic regression and deep neural network models were developed using the aforementioned predictors.

For logistic regression it is important to ensure that the predictors used in the model are not affected by multicollinearity, where the independent variables exhibit a strong correlation with each other (r > 0.9) [44]. However, in our study, we found only a moderate correlation (r = 0.663) between the predictors, which allowed us to proceed with building the model without accounting for this correlation. The specific predictions were made using the following equation:(1)$$p=\frac{1}{1+e^{-z}}$$
where *z* is linear combination of feature $T_{sk}$—average skin temperature and feature $T_{int}$—average internal temperature:(2)$$z=b_{0}+b_{1}*T_{sk}+b_{2}*T_{int}$$

The provided equation can be used to calculate the probability (P) of a patient having pneumonia caused by COVID-19. On the other hand, the probability of being healthy can be calculated as 1-P.

For a deep neural network, the following architecture was chosen, shown in Figure 5. In addition to the input and output layers, four hidden layers were used with additive zero-centered Gaussian noise and dropout to avoid overfitting. The third vertical block on each layer represents the input dimension, fourth—output dimension. “None” represents a dynamic dimension of a batch.

Due to the fact that the dataset is quite small, many computational experiments were carried out using the method of repeated cross-validation to reveal the average accuracy of classification models. The results of computational experiments are shown in Table 4.

The logistic regression model achieved a sensitivity of 97.6% and a specificity of 92.7%. The diagnostic sensitivity of a test measures the percentage of true positives results in all individuals with the studied pathology. The diagnostic specificity of the test shows the proportion of individuals who do not have the test pathologies with negative test results. The diagnostic efficiency—geometric mean of specificity and sensitivity. At the same time, the neural network showed an improvement in sensitivity by 1% and specificity by 7%.

Due to the small dimension of the feature space, the resulting classification models turned out to be not complicated and at the same time highly accurate. Thus, the difference in the efficiency of classification models on training and test datasets does not exceed 1.5% on average.

After training logistic regression on the entire dataset, the following coefficients were obtained for Formula (2):(3)$$b_{0}=-159.613$$
(4)$$b_{1}=1.677$$
(5)$$b_{2}=3.188$$

According to the final model, a statistical evaluation was additionally carried out. The results are shown in Table 5. The model shows high scores and can be applied in practical situations for diagnosing COVID-19 pneumonia.

To visualize the quality of the obtained equation, in Figure 6 shows the results of calculating the probability of having COVID-19 pneumonia in all study participants.

If the value is more than 0.5—COVID-19 pneumonia, if less, then healthy.

## 4. Discussion

MWR has shown reasonable precision in diagnosing pneumonia in COVID-19 patients. However, it is important to note that MWR is not a substitute for other imaging techniques, such as chest CT scans, which remain the primary method for diagnosing COVID-19 pneumonia with complications and have shown the best efficacy. While MWR can be a useful adjunct to other diagnostic methods, it is not yet widely used in clinical practice for the diagnosis of COVID-19 pneumonia. Chest CT scans are preferred since they provide a more detailed view of the lungs and can detect even small areas of lung involvement, including areas that may not be visible on a chest X-ray. It is important to keep in mind that any diagnostic test, including MWR and the chest CT scan, has its limitations, and a diagnosis should not be made based solely on the results of one test. Clinical judgment and other factors, such as a patient’s symptoms, medical history, and other test results, should also be considered when making a diagnosis [16]. A chest CT can be expensive and have limitations such as requiring immobility, and there are alternative imaging methods that may be more suitable in certain situations. One of these alternative methods is a mobile chest X-ray, which can be performed at the bedside and is often used in situations where a patient cannot be moved easily or quickly, such as in an intensive care unit or emergency department. Mobile chest X-ray has its advantages and limitations, however. One advantage is that it is relatively quick and easy to perform and can be carried out with minimal patient movement. It can also be less expensive than chest CT and can be used to monitor changes in lung function over time. However, chest X-rays may not provide the same level of detail as chest CT and may not be able to detect certain types of lung abnormalities or conditions. Today it is possible to use a portable chest X-ray. There are even various artificial intelligence-based systems have been developed for the early prediction of coronavirus using radiography pictures. Additionally, exposure to ionizing radiation from X-rays can be a concern, especially with repeated or prolonged use. In summary, the choice of imaging method depends on the specific clinical situation and should be based on a careful consideration of the benefits and limitations of each method, as well as the risks and benefits to the patient. [45,46]. MWR is a simple and harmless diagnostic test that provides a convenient option for examining the lungs of COVID-19 pneumonia patients, both at the bedside and in other field settings. However, it is essential to note that MWR can only detect temperature differences in the peripheral zones of the lungs, so it may not be effective in investigating deep lung lesions [39]. Conducting MWR studies in patients with complicated COVID-19 pneumonia can provide healthcare providers with diagnostic support to prevent and manage infections associated with this pandemic.

## 5. Conclusions

Microwave radiometry (MWR) is a non-invasive, harmless, and easy-to-use diagnostic tool that has emerged as a promising alternative to chest CT for diagnosing pneumonia in COVID-19 patients. MWR works by measuring the temperature of the lungs, which can be used to identify areas of inflammation or infection.

One of the main advantages of MWR is that it is more convenient than chest CT. MWR can be performed at the bedside or in field settings, and it does not require the use of radiation. This makes MWR a more accessible option for patients in remote areas or in countries with limited healthcare resources.

Another advantage of MWR is that it is more sensitive than chest X-rays. Chest X-rays can miss up to 30% of cases of pneumonia, but MWR has been shown to be more accurate in detecting pneumonia, especially in mild cases.

However, MWR also has some limitations. One limitation is that MWR can only detect temperature differences in the peripheral zones of the lungs. This means that MWR may not be able to detect deep lung lesions. Additionally, MWR is not as accurate as chest CT in diagnosing pneumonia in patients with other medical conditions, such as COPD or asthma.

Despite its limitations, MWR is a promising diagnostic tool for pneumonia in COVID-19 patients. MWR is more convenient and sensitive than chest X-rays, and it does not involve radiation exposure. As a result, MWR may become a more widespread tool in clinical practice in the future.

MWR is still a relatively new technology, and more research is needed to confirm its accuracy and effectiveness.

MWR is not a perfect diagnostic tool, and it should not be used as a substitute for chest CT in all cases.

MWR is a valuable tool for monitoring the condition of patients with severe pneumonia.

Overall, MWR is a promising diagnostic tool for pneumonia in COVID-19 patients. It is more convenient, sensitive, and safe than chest X-rays, and it does not involve radiation exposure. As more research is conducted, MWR may become a more widespread tool in clinical practice.

## Figures and Tables

**Figure 1 diagnostics-13-02585-f001:**
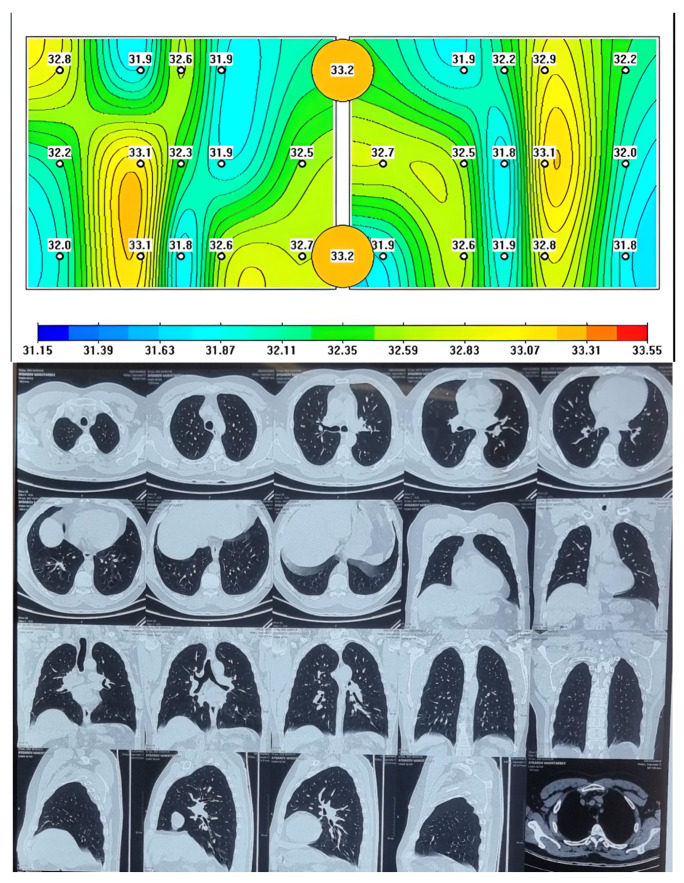
Passive microwave radiometry image and chest CT of a healthy person.

**Figure 2 diagnostics-13-02585-f002:**
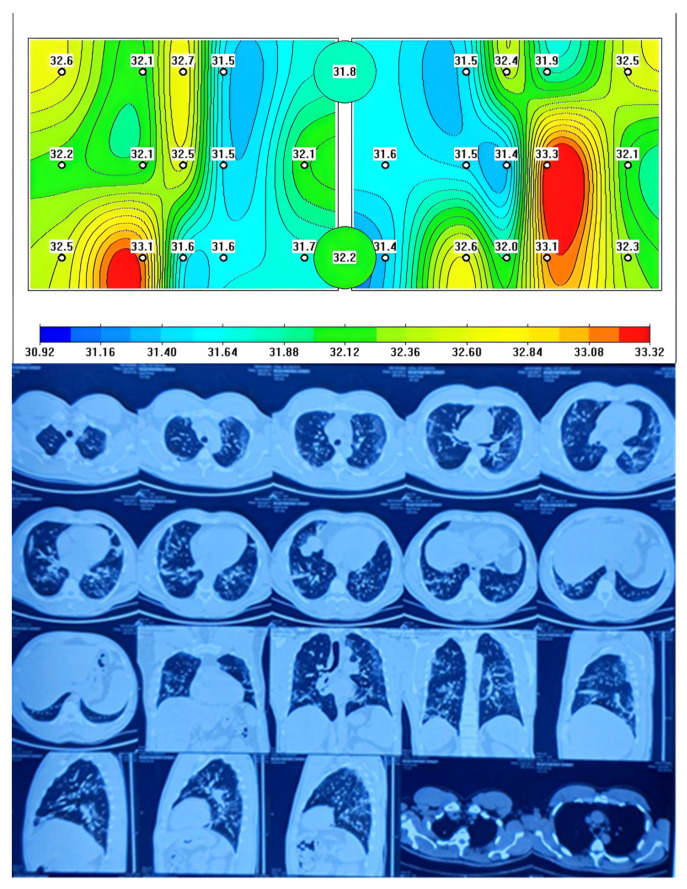
Passive microwave radiometry image and chest CT of a patient with COVID-19 complicated pneumonia.

**Figure 3 diagnostics-13-02585-f003:**
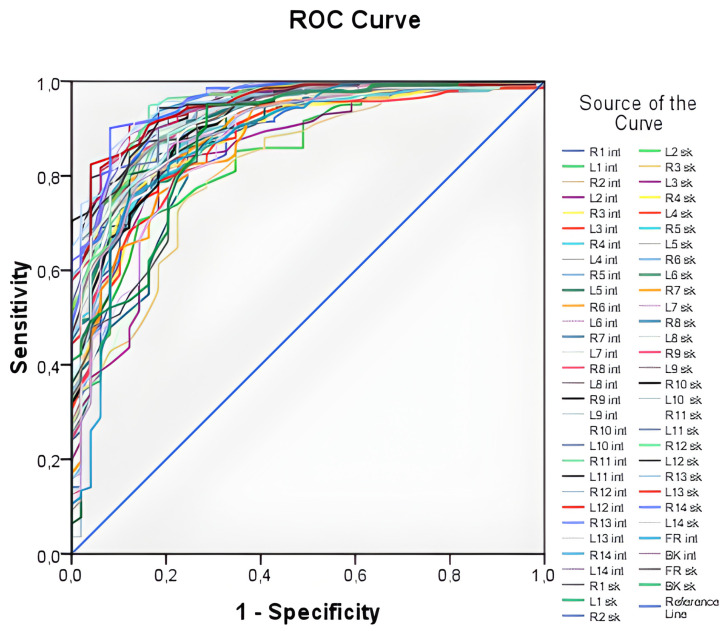
ROC—curves plotted from temperature values at all points under study. The curves generated by the temperature value on the surface of the body (for example, L1_sk_) and the temperature value in the depth of the body (for example, R1_int_) are presented. There is the “1—Specificity”—proportion of incorrectly classified negative cases. “Sensitivity”—proportion of correctly classified positive cases (people with pneumonia).

**Figure 4 diagnostics-13-02585-f004:**
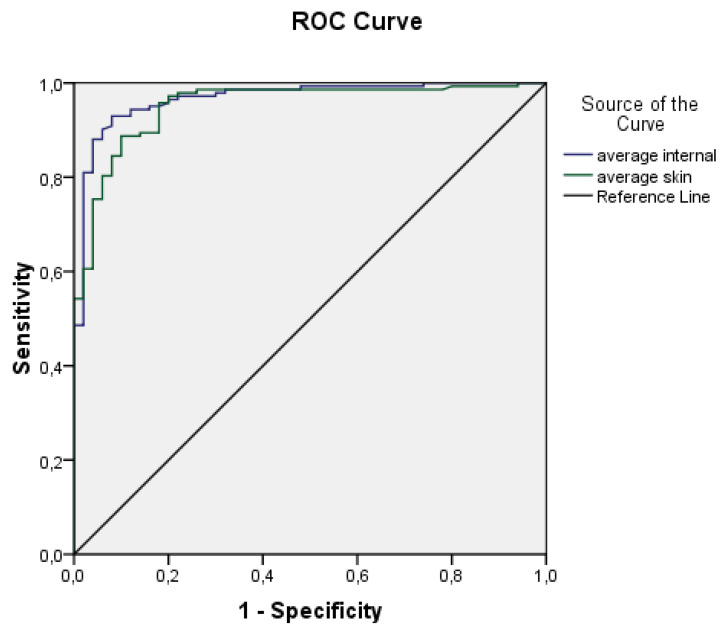
ROC curves plotted from average body surface temperatures (T_sk_) and average core temperatures (T_int_).

**Figure 5 diagnostics-13-02585-f005:**
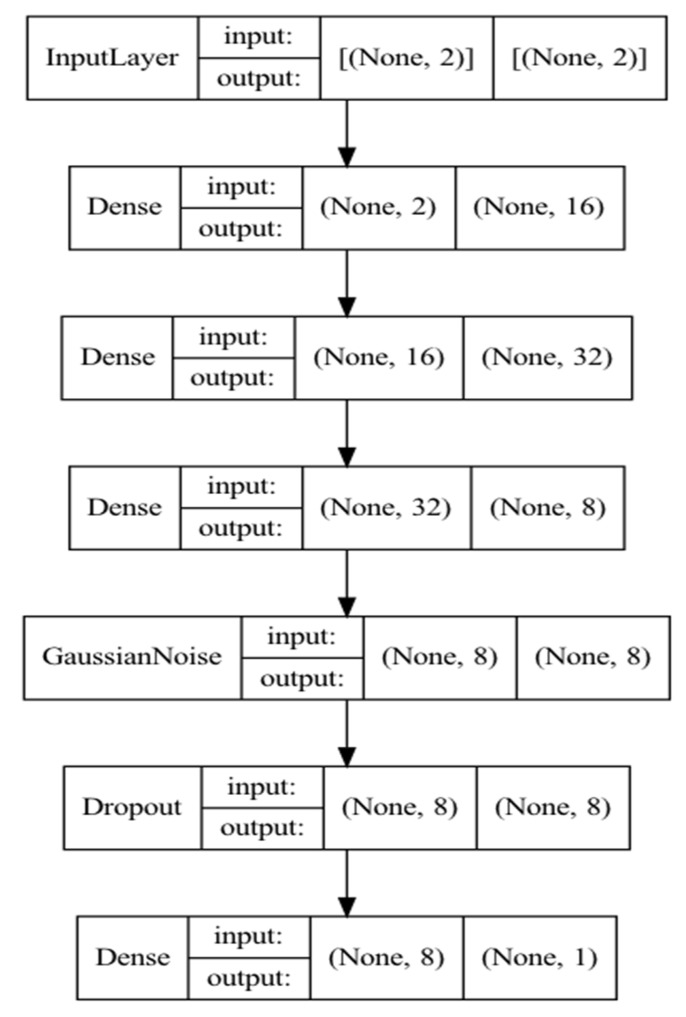
Neural network model architecture.

**Figure 6 diagnostics-13-02585-f006:**
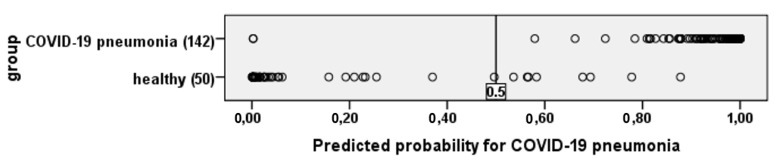
The figure shows that the resulting equation identifies patients with COVID-19 pneumonia and healthy people quite well.

**Table 1 diagnostics-13-02585-t001:** Descriptive Statistics (control group) (in °C).

	N	Minimum	Maximum	Mean	Std. Deviation	95% Confidence Interval	
						Lower	Upper
average internal (T_int_)	50	31.57	34.08	32.60	0.52	32.45	32.74
average skin (T_sk_)	50	29.59	33.06	31.37	0.84	31.13	31.60
difference	50	−0.27	3.16	1.2280	0.73	1.02	1.43

**Table 2 diagnostics-13-02585-t002:** Descriptive Statistics (main group) (in °C).

	N	Minimum	Maximum	Mean	Std. Deviation	95% Confidence Interval	
						Lower	Upper
T_int_	142	32.19	36.85	34.23	0.84	34.09	34.37
T_sk_	142	30.06	36.07	33.21	0.78	33.08	33.34
difference	142	−0.93	3.85	1.02	0.95	0.86	1.18

**Table 3 diagnostics-13-02585-t003:** ROC curves parameters.

Test Result Variable(s).	Area	Std. Error ^a^	Asymptotic 95% Confidence Interval	
			Lower Bound	Upper Bound
average internal (T_int_)	0.967	0.013	0.941	0.993
average skin (T_sk_)	0.951	0.016	0.919	0.983

^a^ Under the nonparametric assumption.

**Table 4 diagnostics-13-02585-t004:** Classification Table.

Observed		ROC Curve Best Thresholds	Predicted Correct	
			Logistic Regression	Deep Neural Network
group	control group	88.8%	92.7%	99.7%
	COVID-19 pneumonia	95.2%	97.6%	98.6%
Overall efficiency		91.5%	94.8%	99.1%

**Table 5 diagnostics-13-02585-t005:** Variable(s): average_internal, average_skin, sex, age. The data was adjusted by gender and age. Cox & Snell R Square = 0.562, Nagelkerke R Square = 0.864, Chi-square for final model 158.71. Hosmer-Lemeshow test, Chi-square = 79,291, *p* = 0.000.

	B	S.E.	Wald	df	Exp(B)	95.0% CI for EXP(B)	
						Lower	Upper
T_int_ (b_1_)	3.188	0.770	17.155	1	24.243	5.363	109.594
T_sk_ (b_2_)	1.677	0.524	10.236	1	5.351	1.915	14.951
Const (b_0_)	−159,463	28,916	30,412	1	0.000		

## Data Availability

Raw data available on request.

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
