# Peer review of "Diagnostic of Patients with COVID-19 Pneumonia Using Passive Medical Microwave Radiometry (MWR)"

_diagnostics, 2023, doi:10.3390/diagnostics13152585_

Round 1
Author Response
Response to Reviewer 1 Comments
Point 1: on the microwave centre frequency and bandwidth
Response 1:
the device operates at 3.5-4.2GhZ with 70MhZ bandwidth
Point 2:
on the wave propagation between antenna and human body. Is there any impedance
match?
Yes. The special mathe mathematical model was developed to insure match [Ref?]
Is the antenna in direct contact with the skin? Yes
Response 2:
yes the sensor is on the surface of the skin during the measurement
Point 3:
on the radiometric calibration and stability
Response 3:
Parameter setting are done during calibration.
Point 4:
on how the infrared and microwave radiometers are used in combination. Is it a single. No
Response 4:
Infrared thermometry is a method for determining skin temperature based on measuring tissue radiation in the infrared wavelength range.. Microwave radiothermometry is a method for measuring the integral deep temperature of human internal tissues based on measuring the power of tissue's own thermal radiation in the microwave wavelength range. The dual band sensor with built-in antenna for simultaneous measurement of core temperature and skin temperature.
Point 5:
Line 100: "Skin and lung temperatures were measured at...": Describe how you determined
these temperatures from the actual radiometer data.
Response 5:
There are 30 points on the participant's chest, 28 of which are symmetrical (R1-R14, L1-L14) and 2 control (FR and BK), configured to perform these measurements, record temperatures and build the corresponding temperature fields. Most of the patients were naked (90) and a minority in thin clothing (52) and the procedure took maximum 10 minutes.
This technique was described in detail in a previous article. We have used formula P=k ∙Trad ∙ ∆f
Point 6:
Line 160: What is "lings temperature"?
Internal Lung temperature
Figures 1 and 2: Spatial correspondence between the microwave and CT images is missing.
Response 6:
Thank you.Corrected.
Since we measured the lungs, there are two areas on the figures. On which we see red areas, which indicates inflammatory areas on both sides. The CT scan also showed bilateral lesions.
Point 7:
Line 166: "with each point recording both skin and lung temperatures", same comment as
on Line 100.
Response 7:
yes, we have removed it.
Point 8:
Line 166: "Inflammation in the form of ground-glass opacity". Could you explain what this
means?
Response 8:
Ground-glass opacity (GGO) is a finding seen on chest x-ray (radiograph) or computed tomography (CT) imaging of the lungs. In the setting of pneumonia, the presence of GGO (as opposed to consolidation) is a useful diagnostic clue. Most bacterial infections lead to lobar consolidation, while atypical pneumonias may cause GGOs. It is important to note that while many of the pulmonary infections listed below may lead to GGOs.
Point 9:
Line 173: "Based on the statement you provided": what do you mean here?
Response 9:
Thank you. We corrected.
Point 10:
Line 185- 186: I do not understand how you formed ROC curves. Please explain in detail and
define what you mean with Sensitivity and with 1-Specificity.
Response 10:
We employed ROC analysis by measured temperatures. This analysis provides accuracy calculation of a binary classification by varying a threshold value T in the range of feature X values. If X < T then the object belongs to healthy group, otherwise to pneumonia group. There is the “1 - Specificity” - proportion of incorrectly classified negative cases. “Sensitivity” – proportion of correctly classified positive cases (people with pneumonia).
Determination of sensitivity and specificity. The diagnostic sensitivity of a test measures the percentage of true positives results in all individuals with the studied pathology. The diagnostic specificity of the test shows the proportion of individuals who do not have the test pathologies with negative test results. 1 - specificity - proportion of incorrectly classified negative cases. In the article, sensitivity refers to the proportion of people with pneumonia, correctly classified by the model, and under specificity, the proportion of people who do not have pneumonia correctly classified by the model. We can agree that the use of these definitions is not entirely correct and replace these concepts. Sensitivity is replaced by - the proportion of people with pneumonia, correct classified by the model (140/142) having pneumonia correctly classified by the model. (42/50)
Point 11:
In the Conclusions Section, please simplify and omit repetitions.
Response 11:
The MWR (microwave radiometry) of the lungs has emerged as a promising diagnostic tool for pneumonia in COVID-19 patients, especially when chest CT is not feasible or unavailable. MWR is non-invasive, harmless, and easy-to-use, making it a more convenient option for examining the lungs of COVID-19 pneumonia patients, both at the bedside and in field settings. While chest CT is considered the gold standard for diagnosing COVID-19 complicated pneumonia and has shown high accuracy, it has some drawbacks such as cost and limited mobility.
However, it is important to note that MWR has its limitations, including its ability to only detect temperature differences in the peripheral zones of the lungs and may not be able to investigate deep lung lesions. Therefore, MWR should not be considered a substitute for chest CT in all cases but rather an alternative when chest CT is not feasible.
Despite its limitations, the study of MWR in patients with COVID-19 complicated pneumonia provides valuable diagnostic support to physicians in preventing and controlling infections related to the pandemic. In addition, MWR is a safer option than chest x-rays, which have been widely used in the past, since it does not involve radiation exposure. Therefore, the use of MWR in clinical practice may become more widespread in the future as a useful tool for diagnosing COVID-19 pneumonia.
Point 12:
Furthermore, the conclusion is a bit contradictive. On the one hand, "MWR is non-invasive,
safe, and efficient" (Line 306-7). But on the other hand, "MWR has its limitations" (Line 313).
How would you like to improve MWR, in order to reduce the limitations.
Response 12:
Perhaps we need to apply it to other conditions like lung cancer, treatment monitoring long term. Lung cancer is a leading cause of cancer-related deaths worldwide, and early detection and effective treatment monitoring are crucial for improving patient outcomes. Current diagnostic methods, such as computed tomography (CT) scans and biopsies, have limitations in terms of invasiveness, cost, and radiation exposure. MWR offers a noninvasive and potentially more accessible approach for monitoring lung cancer.
In the case of lung cancer, the radiometer can be designed to detect and analyze the thermal patterns and characteristics of the lung tissue. Malignant tumors have been shown to exhibit different heat signatures compared to healthy tissue due to increased metabolic activity and altered blood flow. By capturing and analyzing these thermal patterns, medical microwave radiometry can potentially aid in the early detection and monitoring of lung cancer. MWR can help identify subtle thermal changes in lung tissue that may indicate the presence of cancerous growths at an early stage. This could enable early intervention and improve treatment outcomes. During cancer treatment, such as radiation therapy or chemotherapy, medical microwave radiometry can be used to monitor the thermal response of lung tumors. By analyzing the changes in temperature patterns over time, healthcare providers can assess the effectiveness of the treatment and make adjustments if necessary. After completing the primary treatment, regular follow-up monitoring is essential to detect any recurrence or metastasis. MWR can provide a noninvasive and repeatable method for long-term surveillance, allowing for frequent monitoring of lung tissue temperature patterns. By combining medical microwave radiometry with other imaging modalities and biomarkers, healthcare providers can develop personalized treatment plans based on the unique characteristics of each patient's lung cancer. This approach can improve treatment effectiveness and minimize side effects. Compared to invasive procedures such as biopsies or invasive monitoring techniques, MWR offers a noninvasive and painless method for monitoring lung cancer. This can improve patient comfort and compliance with long-term surveillance.

Reviewer 2 Report
This paper presents the Diagnostic of patients with COVID-19 pneumonia using Pas- 2 sive Medical Microwave Radiometry
The literature review needs to be improved (length, comprehension, etc.);
The motivation and original contribution of the work need to be better presented;
A complexity analysis is required;
Discuss the source of images, develop benchmark database
Mention the feature distribution clearly
Measure the complexity of the model
Highlights the major deliverables
Consider related articles as a reference for improving the literature survey, BIFM: Big-Data Driven Intelligent Forecasting Model for COVID-19, COVID-19 diagnosis system by deep learning approaches
Serious proofreading of the manuscript is required.
More Justification is require for the Medical Microwave Radiometry ssyetms in CoVID-19 cases
Need to improve
Author Response
Response to Reviewer 1 Comments
Point 1:
The literature review needs to be improved (length, comprehension, etc.);
Response 1:
The rationale for the use of microwave radiometry (MWR) is the rise of tissue temperature due to the inflammatory process. The local alterations in the synthesis and release of immune mediators in response to injury result in altered biochemical processes and temperature local changes.
Indeed, existing evidence suggests that local inflammation increased temperature that can be detected by the MWR sensor, which is placed on the skin over the inflamed area
The local temperature changes precede structural changes, it is of paramount importance to diagnose inflammatory changes as early as possible, even if signs and symptoms suggestive of inflammation are absent.
Katerina Laskari, Elias Siores,Maria G. Tektonidou, Petros P. Sfikakis. Microwave Radiometry for the Diagnosis and Monitoring of Inflammatory Arthritis. Diagnostics 2023, 13(4), 609; https://doi.org/10.3390/diagnostics13040609
Point 2:
The motivation and original contribution of the work need to be better presented;
Response 2:
The ability of microwave radiometry (MWR) to detect with high accuracy in-depth temperature changes in human tissues is under investigation in various medical fields. There are numerous clinical applications for non-invasive monitoring of deep tissue temperature. We present the design and experimental performance of a MWR measuring volume temperature of tissue regions, especcelly the lungs.
Point 3:
A complexity analysis is required;
Response 3:
Due to the small dimension of the feature space, the resulting classification models turned out to be not complicated and at the same time highly accurate.
Point 4:
Discuss the source of images, develop benchmark database
Response 4:
CT images of the lungs were taken from participants from the main group and MWR images were downloaded in electronic form from the program of the same name
Point 5:
Mention the feature distribution clearly
Response 5:
on MWR images of healthy participants, we can see yellow and green areas. Yellow areas may indicate places with higher temperature values, and green areas with relatively low temperature differences It is important to note that there are no red spots here.
Point 6:
Measure the complexity of the model
Response 6:
The difference in the efficiency of classification models on training and test datasets does not exceed 1.5% on average.
Therefore, we imply that our models are not overcomplex, i.e. there is no overfitting problem. At the same time, the model shows a fairly high classification accuracy.
Point 7:
Highlights the major deliverables
Response 7:
In this study, 142 participants with pneumonia caused by COVID-19 were found to have lung tissue inflammation confirmed in 140 using MWR, of the control group, the absence of inflammation was confirmed in 42 of 50 healthy paricipants.
Point 8:
Consider related articles as a reference for improving the literature survey, BIFM: Big-Data Driven Intelligent Forecasting Model for COVID-19, COVID-19 diagnosis system by deep learning approaches
Response 8:
Today there are even various artificial intelligence-based systems have been developed for the early prediction of coronavirus using radiography pictures.
Yogesh H Bhosale, K Sridhar Patnaik. Application of Deep Learning Techniques in Diagnosis of Covid-19 (Coronavirus): A Systematic Review. Neural Process Lett. 2022 Sep 16;1-53. doi: 10.1007/s11063-022-11023-0.
Point 9:
Serious proofreading of the manuscript is required
Response 9:
We supplemented the introduction of the article, the results and reduced the repetitions in the conclusions, and corrected the erroneous words.
Point 10:
More Justification is require for the Medical Microwave Radiometry ssyetms in CoVID-19 cases
Response 10:
Since the diagnostic method is completely new for detecting pneumonia in patients with COVID-19, it has previously been used once, which we provide in the link.

Round 2
Reviewer 1 Report
Some improvements can be noted. Still there is no information on important properties of the combined IR and MW radiometer: Frequency, bandwidth, polarisation are missing, as well as measures for impedance match at the transition between antenna and human body.
New Tables 1 and 2 seem to be interesting. However there is a lack of a clear definition of what these numbers are, including units.
I am not able to understand the part of the manuscript between Table 4 and Equation (2). I miss definietions of the quantities involved. For me this part is not so relevant. It could be omitted.
Minor errors
Author Response
Open Review
( ) I would not like to sign my review report
(x) I would like to sign my review report
Quality of English Language
( ) I am not qualified to assess the quality of English in this paper
( ) English very difficult to understand/incomprehensible
( ) Extensive editing of English language required
( ) Moderate editing of English language required
(x) Minor editing of English language required
( ) English language fine. No issues detected
Yes |
Can be improved |
Must be improved |
Not applicable |
|
Does the introduction provide sufficient background and include all relevant references? |
(x) |
( ) |
( ) |
( ) |
Are all the cited references relevant to the research? |
( ) |
( ) |
( ) |
( ) |
Is the research design appropriate? |
( ) |
(x) |
( ) |
( ) |
Are the methods adequately described? |
( ) |
( ) |
(x) |
( ) |
Are the results clearly presented? |
( ) |
( ) |
(x) |
( ) |
Are the conclusions supported by the results? |
(x) |
( ) |
( ) |
( ) |
Comments and Suggestions for Authors
- Some improvements can be noted. Still there is no information on important properties of the combined IR and MW radiometer: Frequency, bandwidth, polarisation are missing, as well as measures for impedance match at the transition between antenna and human body.
We have added the following
The lung and skin temperature data were obtained using the MWR2020 device (for-merly RTM-01-RES) MMWR Ltd, Edinburgh, Uk. The MWR2020 operates in the frequency range 3400-3900 MHz. The MWR bandwidth is 500 MHz [46. Vesnin S. G. et al. Portable microwave radiometer for wearable devices //Sensors and Actuators A: Physical. 2021. Vol. 318. PP. 112506.].
The principle of operation of a microwave radiometer is since in the microwave range, the radiation power of human tissues is proportional to the temperature of these tissues. Therefore, by measuring the intrinsic radiation of tissues in the microwave frequency range, it is possible to obtain information about the temperature of internal tissues.
The temperature measured by a microwave radiometer is the average temperature of the tissues in the volume below the antenna. An anti-interference antenna with a slot radiator is used as an antenna. The electric field of the antenna is oriented perpendicular to the slot. The presented antenna allows temperature measurements without additional room shielding. The reflectance of the antenna power for different parts of the body and for different patients is within R2=0 - 0.1. This variation is since the thickness of the fat and muscle layers in different patients can vary significantly. It follows from this that about 10% of the power radiated by human tissues is reflected from the antenna and does not enter the receiving device, which can significantly reduce the accuracy of temperature measurement.
The microwave radiometer compensates for measurement error due to antenna reflections. This is achieved by “noising” the input circuits of the radiometer from the load side of the circulator, which has a temperature close to the measured temperature. Therefore, the microwave radiometer used provides the necessary measurement accu-racy in the presence of antenna mismatch.
- New Tables 1 and 2 seem to be interesting. However there is a lack of a clear definition of what these numbers are, including units.
We have added the units in table 1 and table 2.
- I am not able to understand the part of the manuscript between Table 4 and Equation (2). I miss definietions of the quantities involved. For me this part is not so relevant. It could be omitted.
We have added some corrections in Table 4 and Equation and after that was more clearly, and omitted some text
Comments on the Quality of English Language
Minor errors
Submission Date
21 April 2023
Date of this review
16 Jun 2023 19:56:28

Reviewer 2 Report
Authors not updated all of the given comments in previous round
The revised manuscript is improved but still it has many problem like
Quality of figures is still not as per Journal requirements
Authors have to take their time to improve the overall scientific impact of this paper.
Consider related papers as a reference for improving the literature survey, BIFM: Big-Data Driven Intelligent Forecasting Model for COVID-19 COVID-19 diagnosis system by deep learning approaches
Highlights the major contributions point wise in the introduction section
Show the feature distributions graphically
The language of the paper needs to be improved by the native english speaker
Author Response
Open Review
( ) I would not like to sign my review report
(x) I would like to sign my review report
Quality of English Language
( ) I am not qualified to assess the quality of English in this paper
( ) English very difficult to understand/incomprehensible
( ) Extensive editing of English language required
(x) Moderate editing of English language required
( ) Minor editing of English language required
( ) English language fine. No issues detected
Yes |
Can be improved |
Must be improved |
Not applicable |
|
Does the introduction provide sufficient background and include all relevant references? |
( ) |
( ) |
(x) |
( ) |
Are all the cited references relevant to the research? |
( ) |
( ) |
(x) |
( ) |
Is the research design appropriate? |
( ) |
( ) |
(x) |
( ) |
Are the methods adequately described? |
( ) |
(x) |
( ) |
( ) |
Are the results clearly presented? |
( ) |
(x) |
( ) |
( ) |
Are the conclusions supported by the results? |
( ) |
(x) |
( ) |
( ) |
Comments and Suggestions for Authors
Authors not updated all of the given comments in previous round
The revised manuscript is improved but still it has many problem like
1. Quality of figures is still not as per Journal requirements Authors have to take their time to improve the overall scientific impact of this paper.
We have improved the quality of the pictures. We have added
The MWR of the lungs has emerged as a promising diagnostic tool for pneumo-nia in COVID-19 patients, especially when chest CT is not feasible or unavailable. MWR is non-invasive, harmless, and easy-to-use, making it a more convenient option for examining the lungs of COVID-19 pneumonia patients, both at the bedside and in field settings.
MWR in patients with COVID-19 complicated pneumonia provides valuable diagnostic support to physicians in preventing and controlling infections related to the pandemic. In addition, MWR is a safer option than chest x-rays, which have been widely used in the past, since it does not involve radia-tion exposure. The MWR method can also be used to monitoring the condition of a patient with severe pneumonia, since there is no risk of exceeding the radiation dose. Therefore, the use of MWR in clinical practice may become more widespread in the future as a useful tool, complementing instrumental diagnostic method in patients with COVID-19 pneumonia.
We have revised the Conclusions
Microwave radiometry (MWR) is a non-invasive, harmless, and easy-to-use diagnostic tool that has emerged as a promising alternative to chest CT for diagnosing pneumonia in COVID-19 patients. MWR works by measuring the temperature of the lungs, which can be used to identify areas of inflammation or infection.
One of the main advantages of MWR is that it is more convenient than chest CT. MWR can be performed at the bedside or in field settings, and it does not require the use of radiation. This makes MWR a more accessible option for patients in remote areas or in countries with limited healthcare resources.
Another advantage of MWR is that it is more sensitive than chest x-rays. Chest x-rays can miss up to 30% of cases of pneumonia, but MWR has been shown to be more accurate in detecting pneumonia, especially in mild cases.
However, MWR also has some limitations. One limitation is that MWR can only detect temperature differences in the peripheral zones of the lungs. This means that MWR may not be able to detect deep lung lesions. Additionally, MWR is not as accurate as chest CT in diagnosing pneumonia in patients with other medical conditions, such as COPD or asthma.
Despite its limitations, MWR is a promising diagnostic tool for pneumonia in COVID-19 patients. MWR is more convenient and sensitive than chest x-rays, and it does not involve radiation exposure. As a result, MWR may become a more widespread tool in clinical practice in the future.
MWR is still a relatively new technology, and more research is needed to confirm its accuracy and effectiveness.
MWR is not a perfect diagnostic tool, and it should not be used as a substitute for chest CT in all cases.
MWR is a valuable tool for monitoring the condition of patients with severe pneumonia.
Overall, MWR is a promising diagnostic tool for pneumonia in COVID-19 patients. It is more convenient, sensitive, and safe than chest x-rays, and it does not involve radiation exposure. As more research is conducted, MWR may become a more widespread tool in clinical practice.
- Consider related papers as a reference for improving the literature survey, BIFM: Big-Data Driven Intelligent Forecasting Model for COVID-19 COVID-19 diagnosis system by deep learning approaches
We have added the references according BIFM (13. Dash S., Chakraborty C., Giri S. K., Pani S. K. and Frnda J., BIFM: Big-Data Driven Intelligent Forecasting Model for COVID-19. IEEE Access, vol. 9, pp. 97505-97517, 2021, doi: 10.1109/ACCESS.2021.3094658. 30. Bhuyan, H. K., Chakraborty, C., Shelke, Y., & Pani, S. K. (2022). COVID-19 diagnosis system by deep learning approaches. Expert Systems, 39( 3), e12776. https://doi.org/10.1111/exsy.12776.)
- Highlights the major contributions point wise in the introduction section.
Show the feature distributions graphically
We have revised and added some changes and we have added the feature distributions graphically (Figure 6.The figure shows that the resulting equation identifies patients with COVID-19 pneumonia and healthy people quite well)
Comments on the Quality of English Language
- The language of the paper needs to be improved by the native english speaker
Language was checked
Submission Date
21 April 2023
Date of this review
- un 2023 16:53:13

Round 3
Reviewer 1 Report
N.A.
Line 83: Change "country like" to "countries like"
and check for similar typos.
Author Response
Review Report Form
Open Review
( ) I would not like to sign my review report
(x) I would like to sign my review report
Quality of English Language
( ) I am not qualified to assess the quality of English in this paper
( ) English very difficult to understand/incomprehensible
( ) Extensive editing of English language required
( ) Moderate editing of English language required
(x) Minor editing of English language required
( ) English language fine. No issues detected
Yes |
Can be improved |
Must be improved |
Not applicable |
|
Does the introduction provide sufficient background and include all relevant references? |
(x) |
( ) |
( ) |
( ) |
Are all the cited references relevant to the research? |
(x) |
( ) |
( ) |
( ) |
Is the research design appropriate? |
(x) |
( ) |
( ) |
( ) |
Are the methods adequately described? |
(x) |
( ) |
( ) |
( ) |
Are the results clearly presented? |
(x) |
( ) |
( ) |
( ) |
Are the conclusions supported by the results? |
(x) |
( ) |
( ) |
( ) |
Comments and Suggestions for Authors
N.A.
Comments on the Quality of English Language
- Line 83: Change "country like" to "countries like"
- and check for similar typos.
Submission Date
21 April 2023
Date of this review
07 Jul 2023 21:57:15
We have checked all spelling errors. See the list below
- Line 83
- in orderin introduction on page 2
- the tissue's in introduction on page 2
- a built-in in introduction on page 2
- own radio-thermal
- an MWR introduction on page 3
- the volume introduction on page 3
- were in thin clothing materials and methods on page 3
- frequency range of 3400-3900 MHz in materials and methods on page 3
- levels in medical the rooms in materials and methods on page 4
- of the Medical Center of Kyrgyz State Medical Academy for the in results on page 4
- of the a healthy person in results on page 5
- The Figure 2 shows an MWR in results in page 6
- temperature differences were observed in results in page 6
- provides an accuracy in results in page 6
- to the healthy group, otherwise to the pneumonia group in results in page 7
- a logistic regression in results in page 9
- in the model in results in page 9
- patient having pneumonia in results in page 10
- the input in results in page 10
- achieved a sensitivity in results in page 11
- on a chest X-ray in discussions and conclusions on page 12
